# NF-κB Signaling Is Regulated by Fucosylation in Metastatic Breast Cancer Cells

**DOI:** 10.3390/biomedicines8120600

**Published:** 2020-12-12

**Authors:** Emma H. Doud, Trupti Shetty, Melissa Abt, Amber L. Mosley, Timothy W. Corson, Anand Mehta, Elizabeth S. Yeh

**Affiliations:** 1Department of Biochemistry and Molecular Biology, Indiana University School of Medicine, Indianapolis, IN 46202, USA; edoud@iu.edu (E.H.D.); almosley@iu.edu (A.L.M.); tcorson@iu.edu (T.W.C.); 2Department of Ophthalmology, Indiana University School of Medicine, Indianapolis, IN 46202, USA; tshetty@iu.edu; 3Department of Pharmacology and Toxicology, Indiana University School of Medicine, Indianapolis, IN 46202, USA; mabt@iu.edu; 4Center for Computational Biology and Bioinformatics, Indiana University School of Medicine, Indianapolis, IN 46202, USA; 5Melvin and Bren Simon Comprehensive Cancer Center, Indiana University School of Medicine, Indianapolis, IN 46202, USA; 6Department of Cell and Molecular Pharmacology and Experimental Therapeutics, Medical University of South Carolina, Charleston, SC 29425, USA; mehtaa@musc.edu

**Keywords:** fucosylation, N-glycosylation, breast cancer, metastasis, NF-κB

## Abstract

A growing body of evidence indicates that the levels of fucosylation correlate with breast cancer progression and contribute to metastatic disease. However, very little is known about the signaling and functional outcomes that are driven by fucosylation. We performed a global proteomic analysis of 4T1 metastatic mammary tumor cells in the presence and absence of a fucosylation inhibitor, 2-fluorofucose (2FF). Of significant interest, pathway analysis based on our results revealed a reduction in the NF-κB and TNF signaling pathways, which regulate the inflammatory response. NF-κB is a transcription factor that is pro-tumorigenic and a prime target in human cancer. We validated our results, confirming that treatment of 4T1 cells with 2FF led to a decrease in NF-κB activity through increased IκBα. Based on these observations, we conclude that fucosylation is an important post-translational modification that governs breast cancer cell signaling.

## 1. Introduction

Fucosylation is a type of N-linked glycosylation defined as the addition of fucose sugars in a branched structure to a carrier protein. Although it has been suggested as a putative biomarker in pancreatic and hepatocellular carcinoma [1,2,3,4,5,6,7,8], research linking breast cancer prognosis to N-glycosylation is at an early stage [9]. In mammals, two distinct metabolic pathways supply L-fucose, i.e., a de novo synthesis pathway and a salvage pathway that both rely on the transport of sugars (i.e., mannose, glucose, and fucose) into the cell. Once synthesized or transported in the cell, L-fucose is converted to GDP-L-fucose and attached to proteins within the Golgi–endoplasmic reticulum (ER).

Evidence suggests that being able to target or utilize glyco-modifications and the enzymes that regulate this process has significant clinical potential for cancer [10,11]. Global changes in glycosylation can be detected between samples from breast cancer patients, when compared to those from non-cancer populations, and are associated with metastatic progression of breast cancer [7,12,13,14,15,16,17]. Increased fucosylation is found in invasive breast cancer cell lines and patient cohorts with aggressive breast cancer [9,13,18]. Several studies show that treatment of 4T1 metastatic mammary tumor cells with a fucosylation inhibitor reduces the migratory and invasive qualities of these cells [9,19,20,21]. However, fucosylation signaling in cancer needs further investigation.

In the present study, we performed tandem-mass-tag (TMT) proteomics on 4T1 metastatic mammary tumor cells treated with a fucosylation inhibitor, 2-fluorofucose (2FF) or vehicle (DMSO) control. Over 5000 proteins were quantified, and over 400 proteins were significantly changed in the 2FF-treated samples compared to the DMSO-treated samples. While the proteins that increased in abundance were involved in spliceosome, ribosome biogenesis, and DNA replication, proteins that decreased in abundance upon treatment with 2FF highlighted the NF-κB and TNF signaling pathways, as well as membrane and vesicle-mediated transport. NF-κB is of particular interest because it is a transcription factor that is pro-tumorigenic and a prime target in human cancer. We used traditional proteomic and signaling pathway analysis—western blotting and transcript analysis—to validate our findings. This validation not only confirmed our pathway analysis but also demonstrated the sensitivity of our proteomic methodology.

## 2. Experimental Section

### 2.1. Cell Culture

#### (Full Details for Reagents and Global Proteomics in Appendix A) 

We purchased 4T1 cells from ATCC^®^ (CRL-2539™). Cells were maintained in a humidified 5% CO_2_ incubator at 37 °C and grown in RPMI 1640 medium (Corning, Corning, NY, USA) supplemented with 10% heat-inactivated Fetal Bovine Serum (FBS) (Gibco, Thermo Fisher Scientific, Waltham, MA, USA), L-Glutamine (Corning, Corning, NY, USA), and a Penicillin–Streptomycin solution (Corning, Corning, NY, USA); 2-deoxy-2-fluoro-L-fucose (2FF) was purchased from Synthose Inc. (Concord, ON, Canada); 4T1 cells were treated with 500 µM 2FF for 48 h prior to analysis.

### 2.2. Protein Analysis

Cells were lysed in a buffer containing 50 mM Tris-HCl, pH 7.5, 150 mM NaCl_2_, 1 mM EDTA, 1% Triton X-100 with the HALT protease and phosphatase inhibitor cocktail (Thermo Fisher Scientific, Waltham, MA, USA). Western and lectin blot analysis were performed on the Protein Simple FluorChem-R imaging system.

### 2.3. RNA Analysis

RNA was isolated from cells using the GeneJet RNA isolation kit (Thermo Fisher Scientific, Waltham, MA, USA), and cDNA was synthesized using BioRad iScript Supermix. Reactions were run on either a StepOne or a ViiA7 thermal cycler (Applied Biosystems, Thermo Fisher Scientific, Waltham, MA, USA). mRNA expression was quantified using a standard curve or the ΔΔC_t_ method, normalized to the expression levels of *Gapdh*, *Hprt*, or *Tbp*, and compared to controls.

### 2.4. Quantitative Global Proteomic Comparison of Protein Levels

Sample preparation, mass spectrometry analysis, bioinformatics, and data evaluation were performed in collaboration with the Proteomics Core Facility at the Indiana University School of Medicine (IUSM) [22,23,24,25]. The mass spectrometry proteomics data have been deposited in the ProteomeXchange Consortium via the PRIDE partner repository, with the dataset identifier PXD021413 and 10.6019/PXD021413 [26].

## 3. Results

### 3.1. Pharmacological Inhibition of Fucosylation Alters N-Glycan Processing

To identify signaling nodes that are regulated by fucosylation, we treated 4T1 metastatic mammary tumor cells with the fucosylation inhibitor, 2FF, or DMSO (Appendix A) and performed TMT proteomics (Appendix A) [27]. This analysis identified 5750 proteins and quantified 5288 of them by MS2-based TMT. Over 400 proteins showed significant changes (abundance ratio *p*-value ≤ 0.05) in 2FF-treated samples compared to DMSO-treated samples (Appendix A).

Pathway analysis using STRING.db, Gene Ontology (GO), Kyoto Encyclopedia of Genes and Genomes (KEGG), and Reactome [28,29,30,31] highlighted several cancer-related pathways as downregulated in 2FF-treated cells, in contrast with DNA/RNA processing pathways which were upregulated (Table 1). In particular, we saw decreases in the abundance of proteins that participate in the calnexin/calreticulin cycle, N-glycan trimming in the ER, ER-to-Golgi anterograde and retrograde transport, transport and subsequent modification of glycans in the Golgi, and translocation of glucose transporter 4 (GLUT4) to the plasma membrane. Several key proteins within the glycan synthesis pathway were significantly altered in our 2FF-treated samples (Table 1; Figure 1, *p* ≤ 0.05). Our results showed that the levels of an upstream enzyme of this pathway, hexokinase-2, were significantly higher in DMSO-treated samples compared with 2FF-treated samples. This enzyme converts d-glucose to α-d-glucose-6-phosphate, an early step in glycolysis and de novo biosynthesis of monosaccharides used for N-glycan processing [32]. However, we also saw a reduction in 2FF-treated samples of 15 enzymes involved in asparagine N-linked glycosylation and of 6 major enzymes downstream of hexokinase that generate monosaccharides for N-glycan processing [32].

### 3.2. NF-κB Activity Is Reduced in 2FF-Treated 4T1 Cells

STRING-DB.org pathway analysis (Table 1) allowed us to pinpoint multiple proteins decreasing in abundance clustered around the pathways regulating nuclear factor κB (NF-κB), including tumor necrosis factor (TNF) and toll-like receptor (TLR) signaling. This finding was supported by KEGG pathway analysis, which showed that our proteomic analysis identified at least 11 different proteins which decreased significantly in abundance in response to 2FF treatment within the NF-κB and TNF signaling pathways, which have significant overlap (Figure 2A and Appendix A). 

To highlight the sensitivity and precision of our semi-quantitative proteomic findings, we focused on two specific NF-κB upstream signaling effectors that showed relatively low abundance changes, Tollip and Bcl10 (Figure 2B, red dots), which were among 11 proteins across the NF-κB, TNF, and TLR pathways with NF-κB as a central mechanism in our KEGG and Reactome analyses (Figure 2A, Appendix A) [33]. We directed our analysis to these two proteins based on low abundance changes to highlight the technical sensitivity of our TMT–LC–MS technique. Functionally, Tollip and Bcl10 are linked to the regulation of pro-inflammatory responses through the activation of NF-κB, a transcription factor that is pro-tumorigenic and a prime target in human cancer [34,35,36]. Furthermore, prior studies showed that these pathways are important for the progression of cancer, including breast cancer ([37,38,39,40]. NF-κB has been implicated in the development of hormone-independent, invasive, high-grade, and late-stage breast cancer phenotypes [41]. To validate the reduction in these proteins that were identified in the TMT proteomic analysis, we used western blot to probe for protein expression levels in DMSO- or 2FF-treated 4T1 cells (Figure 3A). The western blot analysis confirmed that Tollip and Bcl10 were significantly decreased in the 2FF-treated 4T1 cells. Furthermore, the decrease in protein expression of these two factors was not due to transcriptional downregulation, as we found no difference in the transcript levels of either *Tollip* or *Bcl10* (Figure 3B).

To determine if the reduced abundance of Tollip and Bcl10 resulted in a decrease in NF-κB expression or activity, we performed western blotting to probe for changes in NF-κB levels and phosphorylation. We treated 4T1 cells with either DMSO or 2FF for 48 hrs. Using an antibody that detects the phosphorylation of the NF-κB subunit p65 at serine 536 (pS536), we found that the phosphorylation of NF-κB was decreased but that NF-κB levels were unaltered (Figure 3C). The regulation of NF-κB occurs through I kappa B alpha (IκBα), which inhibits NF-κB transcriptional activity. Since IκBα was not identified in our global proteomics experiment, we probed for its expression in DMSO- and 2FF-treated 4T1 cells using an antibody. We found that IκBα expression was elevated in 2FF-treated cells compared to DMSO-treated cells, consistent with our finding that NF-κB was less active in these cells (Figure 3C).

NF-κB is a transcription factor; therefore, to further validate these findings, we quantified the expression of NF-κB target genes. We found a decrease of NF-κB-regulated genes that affect angiogenesis, a process that is critical for metastasis. Specifically, we saw decreases in *Icam1* and *Tnfa* (Figure 3D). TNF-alpha activates TNF receptor 1 (TNFR1), leading to the activation of NF-κB transcriptional regulation of *Icam1* [42,43,44,45]. This finding is consistent with the pathway analysis of global proteomic changes, which showed a reduction in TNF and TNFR1-induced NF-κB signaling (Table 1 and Figure 3).

## 4. Discussion

Fucosylation, a post-translational modification that regulates intracellular signaling, is poorly understood. In this study, we performed TMT proteomics to identify signaling pathways affected by fucosylation. The results of the pathway analysis of our global proteomics data identified 11 members of the NF-κB and TLR pathways whose levels were significantly decreased. We chose two upstream regulators of NF-κB with a low change in abundance to highlight the innate sensitivity of this type of proteomic analysis and validate this analysis.

In consideration of our validated targets, Tollip and Bcl10, the role of Tollip in breast cancer has not been extensively explored, making our present study highly novel. Tollip is an adaptor protein that interacts with target of Myb protein 1 (TOM1), TLRs, and interleukin-1 receptor accessory protein (IL1RAP) [46,47,48]. These factors, and in particular TLRs, are readily expressed in breast cancer cells and likely serve an important function in coordinating tumor and immune cell interactions within the tumor microenvironment [49,50]. Of note, we identified one unique peptide from TLR2, a receptor upstream of Tollip (see Figure 3A), that was marginally outside the quality control cutoff for quantification and reporting. This observation indicates that a deeper proteome fractionation will likely yield further signaling insights. Although the upstream regulation of Bcl10 via receptor activation is less well understood, recent advances have demonstrated that Bcl10 is associated with the caspase recruitment domain (CARD)-and membrane-associated guanylate kinase-like domain-containing protein (CARMA-3) and mucosa-associated lymphoid tissue translocation protein 1 (MALTA) [51,52,53,54]. This complex, consisting of CARMA-3, MALTA, and Bcl10, is downstream of epidermal growth factor receptor (EGFR) and is required for EGFR-mediated activation of NF-κB [52]. EGFR was previously shown to be fucosylated, indicating a functional role for fucosylation in EGFR signaling [55,56]. Interestingly, CARMA3 (aka CARD10) was also found in our study to be decreased upon 2FF treatment (Appendix A).

Taken together, these results point towards fucosylation as a regulator of the inflammatory response in metastatic breast cancer. We note limitations exist to quantitation by either Western blot or discovery proteomics—for instance, antibodies commonly utilized may lack specificity or not recognize various protein epitopes, whereas mass spectrometry is still limited by speed, ability to ionize peptides/proteins of interest, and need to separate species prior to ion detection. Future work will focus on how post-translational modifications are affected by fucose inhibition and on the identification of fucose carrier proteins and of potential druggable targets in invasive metastatic breast cancer.

## Figures and Tables

**Figure 1 biomedicines-08-00600-f001:**
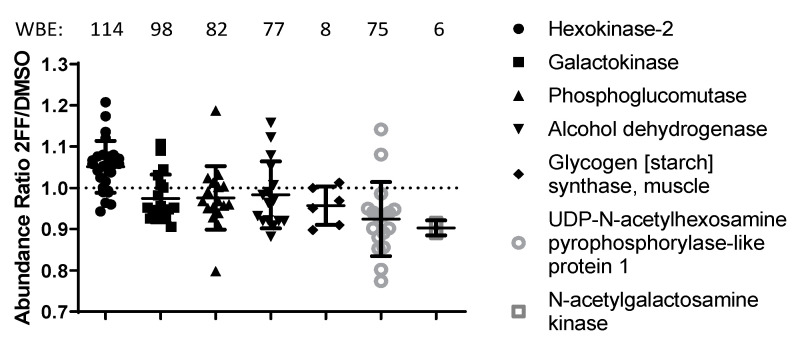
Galactose catabolism and N-glycan trimming in the the ER is altered in cells treated with the fucosylation inhibitor. Graph showing the abundance ratios of individual peptides quantified for select proteins of carbohydrate pathways that significantly change in abundance. For each protein, the western blot equivalents (WBE) or total peptidespectrum matches (PSMs) is shown.

**Figure 2 biomedicines-08-00600-f002:**
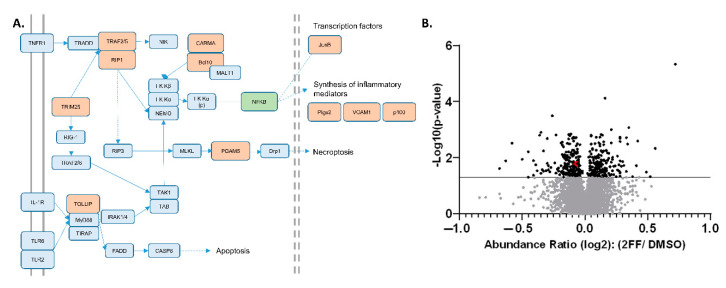
Significantly decreasing proteins associated with NF-κB and TNF pathways. (**A**) Pathway analysis; significantly decreased proteins are shown in orange (see Appendix A for source KEGG pathways). (**B**) Volcano plot of over 5000 proteins quantified in the global proteomics experiment. Black points above the solid line indicate proteins with abundance ratio *p*-values ≤ 0.05; with Tollip and Bcl10 are highlighted (red dots, overlapping).

**Figure 3 biomedicines-08-00600-f003:**
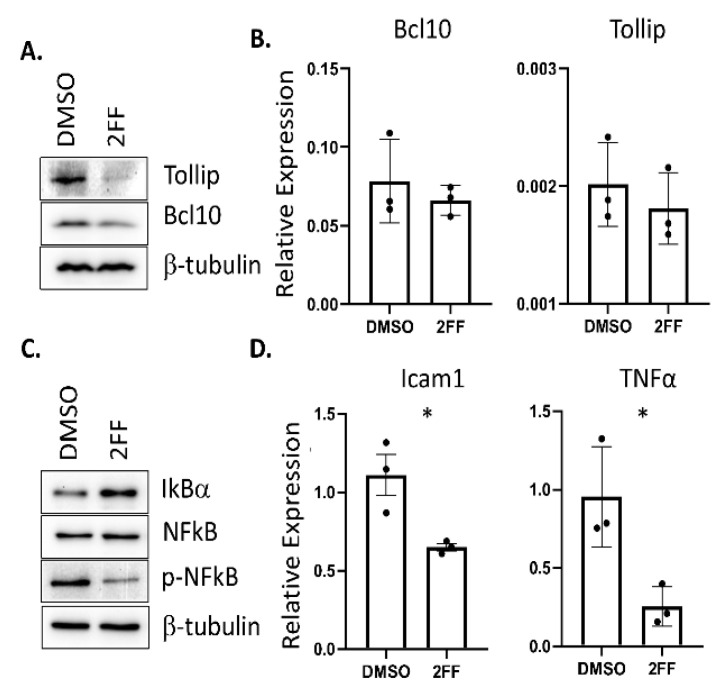
NF-κB activity is reduced in 2FF-treated 4T1 cells. (**A**) Western blots showing a decrease in Tollip and Bcl10 levels after 2FF treatment. (**B**) Quantitative real-time PCR analysis showing no significant change in the transcripts of *Bcl10* and *Tollip* upon treatment with 2FF. (**C**) Western blots showing a decrease in phospho-NF-κB and an increase in IκBα upon 2FF treatment. (**D**) Quantitative real-time PCR analysis showing a decrease in *Icam1* and *Tnfa* upon treatment with 2FF; * *p* < 0.05 for both *Icam1* and *Tnfa*, Student’s *T*-test.

**Table 1 biomedicines-08-00600-t001:** Kyoto Encyclopedia of Genes and Genomes (KEGG) and Reactome pathway analysis. Top pathways with increased or decreased levels of proteins in 2FF- versus DMSO-treated 4T1 cells. FDR calculated through STRING-DB.org. We note that NF-κB and TNF signaling pathways contain common proteins (Appendix A).

KEGG Pathways			Reactome Pathways		
Decreased					
Description	Count in network	FDR	Description	Count in network	FDR
NF-κB signaling pathway	9 or 93	0.0002	Vesicle-mediated transport	23 of 553	1.39 × 10^−5^
RIG-1-like receptor signaling pathway	6 of 68	0.0062	Membrane trafficking	22 of 523	1.39 × 10^−5^
Tight Junction	9 of 165	0.0062	ER to Golgi Anterograde Transport	12 of 147	1.45 × 10^−5^
Protein processing in the ER	9 of 161	0.0062	Asparagine N-linked glycosylation	15 of 269	3.02 × 10^−5^
TNF signaling pathway	7 of 108	0.0068	Immune system	38 of 1523	4.78 × 10^−5^
Increased					
Spliceosome	16 of 130	1.73 × 10^−11^	Metabolism of RNA	33 of 448	4.42 × 10^−18^
Ribosome biogenesis in eukaryotes	7 of 76	5 × 10^−4^	Processing of Capped Intron-containing Pre-mRNA	23 of 212	1.02 × 10^−15^
Huntington’s disease	8 of 187	0.0134	mRNA splicing – Major Pathway	19 of 156	8.94 × 10^−14^
DNA replication	4 of 35	0.0134	Gene Expression (transcription)	30 of 858	7.30 × 10^−9^

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
