# Peer review of "NF-κB Signaling Is Regulated by Fucosylation in Metastatic Breast Cancer Cells"

_biomedicines, 2020, doi:10.3390/biomedicines8120600_

Round 1

Reviewer 1 Report

In this study, the authors performed tandem-mass-tag (TMT) proteomics on 4T1 metastatic mouse breast cancer cells treated with or without a fucosylation inhibitor, 2flurofucose (2FF) or vehicle control. The purpose was to understand the signaling and functional outcomes that are regulated by fucosylation. The analysis identified more than 400 proteins that were significantly increased or decreased in 2FF-treated samples compared to DMSO treated samples. Pathway analysis highlighted several cancerrelated pathways that are downregulated and a number of DNA/RNA processing pathways that were upregulated. The authors mainly focused on the immune regulatory pathways and specifically found that the NF-kB, TNF, and TLR signaling pathways were downregulated by 2FF treatment. Related to NFkB activation/signaling, the authors verified that Tollip, Bcl10, Icarn1, and TNFalpha are reduced by 2FF treatment.

This study highlights the regulation of NFkB by fucosylation in breast cancer cells. However, the study is limited to one cell line, and the findings are descriptive—there is no mechanistic insight as per how fucosylation directly regulates NFkB signaling in this context. There is also no mechanistic insight connecting Tollip, Bcl10, and the signaling pathway that the authors focus on. The experimental design does not delineate directly fucosylated proteins from those that are indirectly affected by fucosylation. This is a large unknown throughout the manuscript. Furthermore, the relevance of this signaling to the biology of these breast cancer cells is unclear. Thus, although this study provides novel proteomic profiling insight, the level of scientific and biological advance and significance is limited.

Specific comments:
1) FIG 1A: Blot is not sufficiently good quality and the labeling tickmarks are inconsistence between 2 blots. In general, the quality of all of figures could use some professional improvement.

2) FIG 3A: The volcano plot reflects that Tollip and Bcl10 are clearly not the most downregulated proteins in 2FF. There is no clear rationale for which the authors choose these 2 proteins; they should provide stronger rationale.

3) FIG 3B: Based on a string interaction, the authors claim that Tollip and Bcl10 are in common pathways with NF-kB, TNF and TLR, which may not be true in their system. Furthermore, Tollip is not associated in any of the pathways that Bcl10 isl; the authors do not address the role it may be playing and how. Thus, what is the rationale for focusing on Tollip?

4) String is purely based on a huge dataset and the shared pathways observed might be in totally different system and different biological conditions. In claiming the involvement of these two proteins, the authors should at least genetically manipulate the proteins ± 2FF to show their contribution to NFkB signaling.

5) FIG 4C: the authors do not show if Tollip or Bcl10 are altered at the mRNA level through qPCR. Thus, it is unclear what effect (whether direct or indirect) fucosylation might be having on the transcriptional or protein level.

6) 4T1 cells are a mouse metastatic breast cancer cell line. It is unclear why the authors have selected this cell line for analysis. This being the only line that they have evaluated, it is unclear what implications and connections can be drawn to the human scenario. Are the effects conserved in human breast cancer cell lines or patient biopsies?

Author Response

We appreciate the suggestions made by the reviewers and believe the quality of the manuscript is strengthened after addressing the following points:

Reviewer 1

1) FIG 1A: Blot is not sufficiently good quality and the labeling tickmarks are inconsistence between 2 blots. In general, the quality of all of figures could use some professional improvement.

We thank the reviewer for their suggestion to improve the image quality of Figure 1A. We have replaced the image with a higher quality AAL blot.

2) FIG 3A: The volcano plot reflects that Tollip and Bcl10 are clearly not the most downregulated proteins in 2FF. There is no clear rationale for which the authors choose these 2 proteins; they should provide stronger rationale.

The original manuscript contained the following text with emphasis added: “To validate the quantitative proteomic findings and the pathway analysis, we honed in on two specific NF-κB upstream signaling effectors that were shown to have relatively low abundance changes to highlight the sensitivity and precision of our proteomic analysis; Toll interacting protein (Tollip) and B-cell lymphoma/leukemia 10 (Bcl10).”          

The 400 proteins that were identified as having differential abundance values in 2FF vs DMSO treated cells were all analyzed to identify overall pathways being perturbed by the decrease in fucosylation caused by 2FF treatment (shown in Fig 1 and described in Table 1). 

Since mass spectrometry had already identified and quantified these 400 proteins, we chose Bcl10 and Tollip to validate by traditional western blots precisely because they had some of the smallest abundance changes identified.  We wanted to emphasize that a small but significant change measured by TMT-LC-MS is accurate and relevant to understanding of how cells respond to a decrease in fucosylation.

One of the strengths of proteomics in general is that one can quantify the actual actors within a set of cells or tissue – not trying to deduce a potential downstream effect from genetic or transcriptomic information alone.  Thus, a relative change of only 5% abundance could have a significant impact on the overall function of different pathways within the system.

3) FIG 3B: Based on a string interaction, the authors claim that Tollip and Bcl10 are in common pathways with NF-kB, TNF and TLR, which may not be true in their system. Furthermore, Tollip is not associated in any of the pathways that Bcl10 isl; the authors do not address the role it may be playing and how. Thus, what is the rationale for focusing on Tollip?

We thank the reviewer for pointing out that the original Figure 3B was unclear, we have created a modified Figure 3A to replace which we hope will clarify the data and our conclusions. We have also added Supplemental Figures 1-3, downloaded from KEGG directly showing the relevant enriched pathways on which we based Figure 3A. We do not mean to suggest that Tollip and Bcl10 have a signaling relationship to each other, but that both proteins have a signaling relationship to NF-kB (See new Figure 3A, Supplemental File 2: Supplementary Figures 1-3). The point of the original Figure 3B (now Figure 3A) was to visualize all of the proteins (11 total) that were quantified as relatively decreasing upon 2FF treatment and belong to the enriched pathways (Table 1, STRING-DB, KEGG analysis) including TNF and TLR that affect NF-kB.  We have now clarified this figure to show how Tollip and Bcl10 are upstream in NF-kB pathways and added additional references to support how Tollip and Bcl10 are relevant in 4T1 cells and breast cancer specifically.  We have modified the text to emphasize that these proteins were chosen to validate based on their low but significant abundant changes as well as presence in enriched pathways. The fact that our proteomic findings and pathway analysis is validated by our additional experiments (see Figure 4) is significant in that it proves our TMT-LC-MS methodology gave accurate and tangible results.

4) String is purely based on a huge dataset and the shared pathways observed might be in totally different system and different biological conditions. In claiming the involvement of these two proteins, the authors should at least genetically manipulate the proteins ± 2FF to show their contribution to NFkB signaling.

As indicated in point 3, we have removed the STRING diagram. In addition, we have added references to demonstrate historical links between these signaling factors and NF-kB in invasive metastatic breast cancer (References 36-42).  We also further discuss that Tollip and Bcl10 are decreased at the protein only level (point 5 and Figure 4).  Figure 4 also shows that transcripts of known target genes of NF-κB also decrease after 2FF treatment.

5) FIG 4C: the authors do not show if Tollip or Bcl10 are altered at the mRNA level through qPCR. Thus, it is unclear what effect (whether direct or indirect) fucosylation might be having on the transcriptional or protein level.

We thank the reviewer for the suggestion of this experiment. We have now tested for Tollip and Bcl10 expression in our experimental system and do not see changes between DMSO and 2FF treated samples, suggesting that our effect is at the protein level. These data are reported in Figure 4B.

6) 4T1 cells are a mouse metastatic breast cancer cell line. It is unclear why the authors have selected this cell line for analysis. This being the only line that they have evaluated, it is unclear what implications and connections can be drawn to the human scenario. Are the effects conserved in human breast cancer cell lines or patient biopsies?

The 4T1 cell line is considered representative of human stage IV metastatic breast cancer and is a choice cell line for investigators who want to study the tumor microenvironment using in vivo (ie mouse) modeling, which cannot be studied using a human cell line because immunocompromised mice are required for xenograft tumors to take hold. We chose the 4T1 cell line precisely for the purpose of being able to eventually connect our proteomic findings with deeper pathophysiological changes that occur within the tumor microenvironment.

To address concerns whether implications can be connected with human cancer, there are a growing number of publications that point to a significant role for fucosylation (and other glycosylation modifications, ie other sugar types) as being important in breast cancer. We previously published a study (Int J Mol Sci. 20, 2528.), already highlighted within the manuscript text, that shows that there are specific fucosylated glycoforms that are present in 4T1 cells, xenograft derived tumors resulting from 4T1 cells, and human breast cancer samples (~200 tissue samples). 

Reviewer 2 Report

The authors present a semiquantitative proteomic study of 4T1 metastatic breast cancer cells treated with a fucosylation inhibitor (2FF) versus a control. The results are interesting, as is the subject matter and the manuscript deserves publication after some changes. In particular, the results presented are semiquantitative or relative quantitative. In fact, they represent a difference in quantity compared to a control. An absolute quantification is not done. In the text, however, the results are presented as quantitative and this can be deceiving. Better to write semi-quantitative or relatively quantitative results

Author Response

We appreciate the suggestions made by the reviewers and believe the quality of the manuscript is strengthened after addressing the following points:

The authors present a semiquantitative proteomic study of 4T1 metastatic breast cancer cells treated with a fucosylation inhibitor (2FF) versus a control. The results are interesting, as is the subject matter and the manuscript deserves publication after some changes. In particular, the results presented are semiquantitative or relative quantitative. In fact, they represent a difference in quantity compared to a control. An absolute quantification is not done. In the text, however, the results are presented as quantitative and this can be deceiving. Better to write semi-quantitative or relatively quantitative results

Response:

We thank the reviewer for pointing out that our text was not clear. Isobaric mass spectrometry provides only relative levels of quantitation.  We have modified the text in the abstract and the methods to include this terminology.

Round 2

Reviewer 1 Report

The authors have addressed some of the points that I raised. However, their explanation of they they chose to focus on Tollip and Bcl10 is not satisfactory, and their argument for only using 4T1 cells is also insufficient. Why would the authors want to validate the least changed of protein hits? Further, just because the authors previously showed that 4T1 cells express fucosylated proteoglycans does not mean that they reflect the altered protein profiles and the major hits that they have pursued in human cells. The authors need to include more than 1 cell line. 
